# Fifty Years of the National Rabies Control Program in Brazil under the One Health Perspective

**DOI:** 10.3390/pathogens12111342

**Published:** 2023-11-11

**Authors:** Maria Cristina Schneider, Kyung-Duk Min, Phyllis Catharina Romijn, Nelio Batista De Morais, Lucia Montebello, Silene Manrique Rocha, Sofia Sciancalepore, Patricia Najera Hamrick, Wilson Uieda, Volney de Magalhães Câmara, Ronir Raggio Luiz, Albino Belotto

**Affiliations:** 1Department of Global Health, School of Health, Georgetown University, Washington, DC 20007, USA; 2Institute of Collective Health Studies, Federal University of Rio de Janeiro, Rio de Janeiro 24220-900, Brazil; volney@iesc.ufrj.br (V.d.M.C.); ronir@iesc.ufrj.br (R.R.L.); 3PAHO/WHO (Ret.), Washington, DC 20037, USA; patricia.najera@gmail.com; 4College of Veterinary Medicine, Chungbuk National University, Chungbuk 28644, Republic of Korea; kdmin11@hotmail.com; 5PESAGRO-RIO, Niteroi, Rio de Janeiro 24120-191, Brazil; phyllisromijn@gmail.com; 6Fortaleza Municipal Health Department, Fortaleza, Ceará 60025-000, Brazil; neliobmo@uol.com.br; 7Secretaria de Vigilancia em Saúde e Ambiente, Ministry of Health, Brasilia 70723-040, Brazil; lucia.montebello@gmail.com (L.M.); silene.rocha@saude.gov.br (S.M.R.); 8PAHO/WHO, Washington, DC 20037, USA; sofiasciancalepore@gmail.com (S.S.); albino.belotto@gmail.com (A.B.); 9Department of Zoology and Botany, São Paulo State University (Ret.), Sao Paulo 05508-090, Brazil; wilson.uieda@gmail.com; 10FUNASA (Fundacao SESP)/Ministry of Health (Ret.), Brasilia 70070-040, Brazil

**Keywords:** rabies, control, dog, wildlife, Brazil, risk, prevention

## Abstract

In 1973, the National Rabies Program was created in Brazil through an agreement between the Ministry of Health and Agriculture. Since its beginning, it developed integrated action through access to free post-exposure prophylaxis (PEP) for people at risk, dog vaccination campaigns, a joint surveillance system, and awareness. This study aims to describe human rabies in Brazil under the One Health perspective in recent decades, including achievements in the control of dog-mediated cases and challenges in human cases transmitted by wild animals. This paper also explores possible drivers of human rabies in the Northeast Region with half of the cases. The first part of this study was descriptive, presenting data and examples by periods. Statistical analysis was performed in the last period (2010–2022) to explore possible drivers. Dog-mediated human cases decreased from 147 to 0, and dog cases decreased from 4500 to 7. A major challenge is now human cases transmitted by wild animals (bats, non-human primates, and wild canids). Most current human cases occur in municipalities with a tropical and subtropical moist broadleaf forest biome and a Gini index higher than 0.5. In the multivariable analysis, an association with temperature was estimated (OR = 1.739; CI_95%_ = 1.181–2.744), and primary healthcare coverage (OR = 0.947; CI_95%_ = 0.915–0.987) was identified as a protector. It is possible to significantly reduce the number of dog-mediated human rabies cases through the efforts presented. However, Brazil has wildlife variants of the rabies virus circulating. The association of human cases with higher temperatures in the Northeast is a concern with climate change. To reduce human cases transmitted by wild animals, it is important to continue distributing free PEP, especially in remote at-risk areas in the Amazon Region, and to increase awareness.

## 1. Introduction

Rabies is an ancient zoonosis with the highest case fatality rates and is present in all continents except Antarctica [1]. It has been estimated that 59,000 human fatalities occur each year, with 80% of them occurring in rural and lower-income areas [2]. Dogs are responsible for the virus transmission of 90% of human cases worldwide. Most dog-mediated cases take place in Asia (59.6%) and Africa (36.4%) in people who cannot afford post-exposure prophylaxis (PEP) [2,3,4].

The World Health Organization (WHO) currently classifies rabies as a neglected tropical disease (NTD). NTDs constitute a group of 20 conditions that are mainly prevalent in tropical areas, where they mostly affect impoverished communities and disproportionately affect women and children [5]. However, rabies is not neglected in all countries of the world; many in Europe, North America, and Japan have controlled the disease [6]. After an enormous effort in rabies control, countries in Latin America were also able to reduce the number of dog-mediated cases significantly, including Brazil [7,8,9,10,11,12,13,14].

In 2018, a call for action was launched by setting a goal of worldwide zero human–dog-mediated rabies deaths by 2030. For the first time, four organizations, the World Health Organization (WHO), the World Organization for Animal Health (OIE), the Food and Agriculture Organization of the United Nations (FAO), and the Global Alliance for Rabies Control (GARC), joined forces, as the United Against Rabies collaboration, determined to reach this goal [6].

Rabies was one of the first diseases whose control efforts were approached from across different disciplines in the Region of the Americas. At the end of the 1940s, Veterinary Public Health Programs were created by the Centers for Disease Control and Prevention (CDC) in the United States and by the Pan American Health Organization (PAHO/WHO). Veterinarians started to be part of those health teams controlling zoonotic diseases. During the 1970s, many countries in the Americas Region created their own Rabies Program with PAHO support. The rabies surveillance system was implemented at the PAHO in the late 1960s, monitoring cases in humans and animals (dogs, other domestic animals, and wild animals) [15,16].

In 1973, the National Program of Rabies Prophylaxis in Brazil was created through an agreement among the Ministry of Health, Ministry of Agriculture, CEME (*Central de Medicamentos*—government center to purchase vaccines and medications), and the PAHO. Since the inception of this Program, it has had an integrated vision recommending actions for humans, such as post-exposure prophylaxis, and actions for animals, such as dog vaccination. The surveillance system was a joint effort from the Ministry of Health and the Ministry of Agriculture; both sent the rabies cases to one combined bulletin (human, dog, and cat cases by the Ministry of Health and livestock cases by the Ministry of Agriculture; both reported wild animal cases). This integrated surveillance system is still coordinated regionally by the PAHO [17].

The Ministry of Health coordinated the National Program; all State Health Departments implemented these activities at the first subnational level. Eventually, they were implemented at the municipality level as well (second subnational level). National multi-sectorial and multidisciplinary advisory committees were established and remained active for many years, and frequent examples of joint work among sectors at different administrative levels could be described.

It could be suggested that the Brazilian Rabies Program and the activities developed over the course of these five decades followed the perspective of what is now coined as One Health. Many definitions have been created for One Health, most of them implying collaboration and interdependence across disciplines and sectors [18,19,20,21]. It is essential to understand that animals and humans sharing an environment are interdependent and that they are affected by the socioeconomic interests of humans and suffer external pressures. If joined, different disciplines can provide new methods, tools for research, policies, and services to the benefit of humanity and animals while considering the environment for current and future generations [15].

The WHO currently recommends the One Health approach to control NTDs, and the joint guidelines from FAO and OIE included rabies as a priority zoonotic disease [5,22]. Also, several documents presenting strategies and recommendations for the One Health Joint Plan have been published by the quadripartite (WHO, FAO, WOAH, and UNEP) after the COVID-19 pandemic [23].

Analyzing the origin of the rabies virus, Badrane and Tordo (2001) consider that bats are possibly the primary hosts of *Lyssavirus* [24]. Rabies in hematophagous bats were likely present in the Americas when the earliest colonists arrived [25]. Other bats could also transmit the rabies virus; however, the cases in Brazil were likely transmitted via the common vampire bat (*Desmodus rotundus*), as it is known to feed on animals and eventually on people. This bat species is present only in the Americas region, from Argentina to Mexico, including the entire territory of Brazil [26]. It is probable that rabies virus strains that circulated in dogs in the Americas region came from the European colonizer’s dogs; rabies was a disease common in Europe in the XVIII century [27].

The rabies virus, from the genus *Lyssavirus,* can now be distinguished by molecular characterization into several distinct genotypes; these techniques revealed the existence of distinct variants of rabies viruses distributed among different animal species in different regions of the world [2,27]. In Brazil, seven genetic variants were found: variants 1 and 2, isolated from dogs; variant 3, from the vampire bat (*Desmodus rotundus)*; and variants 4 and 6, from insectivorous bats (*Tadarida brasiliensis* and *Lasiurus cinereus)*. Two other news variants found in *Cerdocyon thous* (bush dogs) and *Callithrix jacchus* (white-tufted marmosets) were not detected before [28].

It should be noted that, in Northeastern Brazil, these new variants of the rabies virus, with no antigenic or genetic relationship to any known rabies variant found in bats or land mammals in the Americas, were identified in association with cases of human rabies reported in the state of Ceará, Brazil, in the 1990s transmitted via *Callithrix jacchus,* commonly called “sagüí” in Portuguese [29]. More recently, the animal of public health concern regarding rabies virus transmission in Brazil is the *Cerdocyon thous,* a wild canid that is responsible for human and dog cases in the Northeast Region.

The experience of Brazil in the control of rabies mediated by dogs to eventual cases, as well as the challenges faced with wild animal rabies, could be shared with other regions of the world. This study aims to describe human rabies in Brazil under the One Health perspective in these five decades, including achievements in the control of dog-mediated cases and challenges of the rabies virus transmitted by wild animals. This paper also explores possible drivers of human rabies in the Northeast Region in half of the cases historically.

## 2. Materials and Methods

### 2.1. Study Area

Brazil is the largest country in Latin America, with a geographic area of 8,515,767 km^2^ and a population of 211,755,692, according to the latest census conducted in 2021 by the Brazilian Institute of Geography and Statistics [30], distributed in 5 regions and 26 states and into the Federal District (first subnational level) and 5570 municipalities (second subnational level).

The Southeast is the most populated region as it makes up 42% of Brazil’s total population and has a higher GDP per capita; the North and Northeast Regions, with a larger proportion of rural populations, have a lower GDP per capita. Brazil’s Gini index is one of the highest in the Americas, estimated at 0.544 in 2021; the Northeast Region reports the highest while the South Region reports the lowest [30].

In Brazil, post-exposure prophylaxis (PEP) has been free of charge to the population since the beginning of the Rabies Program in the 1970s. In the 1990s, the Brazilian Unified Health System (SUS) was established and ran from primary care to high-complexity health services countrywide [31]. According to the SUS strategy, the municipalities could have a Family Health Team, formed by a doctor, nurse, technician, or nursing assistant and community health agents; there may also be a dentist and dental hygiene technician, among other professions [32]. As part of SUS, the no-cost-to-public PEP could be administrated at vaccination rooms, hospitals, emergency care units, and different types of health units.

### 2.2. Study Design

#### 2.2.1. Part One—Describing the Five Decades

The first part of this study consists of a retrospective study using historical data from when the Program was created in 1973 to 2022. This information was briefly analyzed to determine the study periods, mostly based on decades. Five periods were defined: Period A from 1973 to 1979; Period B from 1980 to 1989; Period C from 1990 to 1999; Period D from 2000 to 2009; Period E from 2010 to 2022.

The Program was structured during Period A, and no data on the number of human cases by the possible animal of transmission and by state were found before 1980.

The description of each period with the epidemiological situation related to human and dog cases, major activities, achievements, and challenges are presented; also included are examples of tools developed, interdisciplinary working groups, and some special activities that were performed in each period demonstrating the integrated vision among the animal–human–environment or One Health approach of the Program.

#### 2.2.2. Part Two—Exploring Possible Drivers

For the last period (2010 to 2022), the Ministry of Health has human rabies cases available by the transmitting animal at the second subnational level (municipalities); we use this information to examine possible drivers for human rabies cases. A smaller group of variables (i.e., major habitat type or biome, temperature, GDP per capita, Gini index, population density, and number of habitants) were selected to demonstrate the distribution of human cases in the whole country.

Using aggregated data by municipality, a statistical analysis was performed to explore a larger number of variables as possible drivers only for the Northeast Region of Brazil. The reason for this design was as follows: (a) the number of municipalities with human rabies cases is small, and Brazil has 5570 municipalities; (b) the Northeast Region reported 53% of all human cases, and this region is considered a challenge due to the higher number of human cases and its socioeconomic indicators since the creation of the Program; (c) it is the only region where all four variants associated with human cases (AgV2, AgV3, Ag *Callytrix* sp., and Ag wild canids where detected; and (d) many states of Brazil do not continue annual dog mass vaccination campaigns after there is a decrease in the number of cases in dogs, and we include this variable in the analysis.

We described and examined the association of municipalities with cases of human rabies and possible drivers: environmental (major habitat type or biome; temperature, precipitation, and forest loss), socioeconomic (GDP per capita and Gini index), and demographic (population density and percentage of rural population). As covariables, we used dog vaccination coverage and a proxy of access to PEP using the coverage of primary health care by SUS.

### 2.3. Data Collection, Sources, and Definitions

The notification of cases of human rabies is compulsory in Brazil and part of the National Surveillance System (SINAN per its acronym in Portuguese). The number of rabies cases reported to the Ministry of Health aggregated at the municipality level is of open access in the SINAN database [33]. The total number of human (1986–2022) and canine cases of rabies are publicly available on the government website [34]. From 1973 to 1985, the source was [35] based on the Ministry of Health. Genetic variants of human (2010 to 2022) and canine (2014 to 2022) cases by municipality are also available on the Ministry of Health webpages [34]. All sources of information are in Appendix A. Case definitions used by the Ministry of Health can be found in Appendix A.

To complete the historical data and examples of activities developed during the fifty years of existence of the Program, the authors, most of them related to the Program during different periods, searched their theses, publications, unpublished reports, and archives to select the most relevant information for this study.

Dog vaccination coverage by municipality for the Northeast Region was obtained from an internal report of the Ministry of Health and was available only from 2012 to 2017. Municipalities with more than two years of missing information were excluded; an average was estimated for this period (not including the missing values). Population coverage from Family Health Teams was used as a proxy variable for access to PEP. These data were obtained from an open-access report from the Ministry of Health [32], and data were collected from the same years as the dog vaccination coverage; an average was also estimated for this period, and there were no missing values in this variable. Appendix A outlines how the covariables were estimated (coverage from Family Health Teams and dog vaccination coverage) in this study.

The selection of variables as possible drivers was conducted based on a bibliographic review and on the author’s former studies analyzing drivers for other topics in Brazil using previously geo-processed variables [36,37,38]. Groups of variables were created to explore these possible drivers; these variables were either downloaded or created from original sources gathered from various open-access data sources. The environmental variables used as sources were as follows: major habitat type or biome [39,40,41], temperature [42], precipitation [42], and forest less [43]. For the socioeconomic and demographic variables, the source was IBGE [44] (Appendix A).

### 2.4. Data Analysis

For part I, only descriptive figures were generated in Microsoft Excel (Version 2308). For part II, a GIS database was built from scratch, linking all variable groups to the updated official municipal shapefile from the Brazilian Institute of Geography and Statistics (acronym in Portuguese, IBGE), which is the government’s main provider of open-access data and geographic information of the country. The Program used for geoprocessing different groups of variables was QGIS, where geocoded data were joined by municipality to construct the database, including all the potential drivers, as well as the rabies epidemiological data and health coverage.

For the statistical analysis, a binomial logistic regression model was used to estimate the association between the human rabies cases (0 = municipality without cases; 1 = municipality with one or more cases) and several variables considered as possible drivers and two covariables. In the statistical analysis, we used the biome variable as tropical biome vs. non-tropical and temperature and precipitation as continuous variables. A univariate analysis was conducted, followed by a multivariable analysis, and the odds ratio (OR) was estimated with a 95% confidence interval (CI). The statistical analysis was performed using R software (ver. 4.2.2).

## 3. Results

From 1973 to 2022, a total of 2398 cases of human rabies were reported in Brazil. From 1980 to 2022, 1492 human cases were notified to the system; more than half (53.55%) of these instances were reported in the Northeast Region, which has the lowest GDP per capita in the country (Table 1). The states of Maranhão (most dog-mediated) and Pará (most transmitted by bats) notified around 11% of cases. 

Cases were reported to occur by virus transmitted by different animal species; most of the human cases were dog-mediated (75.1%), followed by bats (12.3%) (Figure 1). However, when described by period, 83.8% of the human cases were dog-mediated from 1980 to 1989; the inverse situation was observed in the last period (2010–2022) when 20.0% of rabies cases were dog-mediated. The number of dog-mediated human cases reduced from 147 in 1980 to zero in 2008 (Figure 1) (Appendix A).

From 1980 to 2022, the system was notified of 28,527 canine cases. In 1980, 4500 dog cases were reported, which reduced to seven cases in 2022, indicating a reduction of 98.8% (Figure 2).

Rabies detected in cats was also notified to the system. These data represented around 10% of the number of dog cases recorded in the earlier years of the Program. Yet, in 2015, when the number of dog cases underwent a significant reduction, more instances of rabies in cats became evident. In 2022, the number of feline cases exceeded the number of dog cases; the variant in these cases was found to have most likely originated in bats. Rabies cases in livestock and wildlife were constantly reported to the system; however, they are not the focus of this study.

### 3.1. Part 1

#### 3.1.1. Period A, from 1973 to 1979—Establishing and Implementing the Program

The Program’s activities in this period focused on dog-mediated rabies, with five major actions: (1) decentralized and free-of-charge post-exposure prophylaxis (PEP) for humans; (2) free mass vaccination campaigns for dogs; (3) surveillance of humans and animals; (4) strengthening laboratory diagnostic capacity; and (5) health awareness and education. These activities started in urban areas of larger cities, later including small towns and rural areas; by 1977, all states were involved [16,25,45].

Guidelines were also created based on WHO recommendations, as well as standards for vaccine production in the country. In this period, nerve tissue vaccines were used (Fuenzalida and Palacios type) and replaced by cell culture vaccines in 2003. The government purchased all vaccines for human and canine use from national laboratories, controlled their quality, and distributed them to all states. Per the guidelines outlined by the government and following WHO recommendations, the PEP will consist of a vaccine or vaccine and serum based on the type of exposure.

The integrated surveillance system was established for rabies using compulsory forms for human cases and animal cases (confirmed by lab and clinical cases), and activities for rabies control were reported monthly. This information was sent back to the states as reports, and an annual report was also sent to the PAHO regional surveillance system.

In these first years of the surveillance system, 927 human cases were reported. A large number of dog cases (46,298 dog cases) were reported; however, the criteria for dog case notification were just being implemented, and the high number of dog cases in 1974 (12,475 dog cases) and 1975 (13,710), compared with an average of 4669 during 1975 to 1979, could be a reflection of the adjustment of the reporting system (Appendix A).

#### 3.1.2. Period B, from 1980 to 1989—Reduction in Dog Transmissions

During this period, the main activities previously established by the Program continued. Free PEP was available in most municipalities (86%) in that first decade; annually, around 2,500,000 human doses of human vaccine were distributed, and serum was administered to around 6% of the people who received PEP according to the type of exposure [35,45].

The major difference between rabies control actions in Brazil during this period versus the first one was the strategy change for mass dog vaccination campaigns; before this was house-to-house vaccinations, which changed to one-day mass campaigns using fixed points in the cities. Around ten million dogs were vaccinated annually, a majority of them in these one-day mass campaigns (or around the same day in the entire country). These events were advertised using free media on the radio and TV. To enact these mass vaccination campaigns, the Program received support from various institutions: veterinary services from the agriculture sector, students, militaries, community associations, and others, as well as many staff from the states and municipalities’ health departments. During Period B, the coverage was around 80% [45].

The laboratory capacity for rabies diagnosis was developed; in 1988, a total of 34 laboratories were functional, and around 12,000 samples were sent for rabies diagnosis every year [35]. Another activity developed was outbreak control; if a rabies case was detected, it was of mandatory notification, and “focus control” needed to be developed within 72 h. During this period, it was common to capture street dogs and take them to the Zoonoses Center to wait for rescue; however, if the dog was not rescued, it was euthanized. This activity has not been performed for many years [16].

During this period, several trainings and meetings were held. Two working groups to support the National Rabies Program were created. The first was for scientific advice; it was a transdisciplinary (human medicine, veterinary medicine, virology, epidemiology, immunology, and bat specialist) and intersectoral group (health sector, agriculture, and university) and some of the State Rabies Coordinators. The second group, more for technical advice, was formed by one State Rabies Coordinator representative for each region; most of the educational materials and guidelines were discussed and approved by this group.

This scientific advice group developed several indicators to evaluate the risk of rabies at the municipality level; the main one was related to surveillance reliability. The “silent area” indicator suggested that a surveillance system is considered unreliable if it does not achieve the recommendation to send 0.2% of annual samples from the estimated canine population for laboratory diagnosis of rabies every year. This indicator was in use at all levels for many years and became part of the SUS indicators for rabies control activities. Later, this indicator was reevaluated and recommended at the region level as 0.1% [12,46]. A tool was developed using these indicators and others divided into three groups: (a) epidemiological situation with a major focus on surveillance, (b) control actions related to humans and animals, and (c) socioeconomic conditions (Table 2) [25].

As a result of all these activities directed to humans and animals, at the end of Period B, the number of dog-mediated human cases reduced by 70.1%, and the number of dog cases reduced by 86.8% (Figure 1 and Figure 2).

In this period, human cases as a result of virus transmitted by marmosets (*Callithrix jacchus*), a non-human primate, were registered for the first time in Brazil in the state of Ceará [29]. Human cases through viruses transmitted by bats (probably the common vampire bat *Desmodus rotundus)* were registered every year in different states.

#### 3.1.3. Period C, from 1990 to 1999—Increasing the Number of Human Cases Transmitted by Bats

In 1990, a large outbreak of human rabies cases, as a result of the virus being transmitted by hematophagous bats, occurred in the state of Mato Grosso in the Legal Amazon Region, with eight cases officially reported in a gold mining camp; however, more fatalities likely occurred in this remote area. In total, 11 human cases through virus transmitted by bats were reported that year, and in 1992, another 13 cases were reported. These events brought awareness to the rabies virus cycle, transmitted by the common vampire bat (*Desmodus rotundus)* and threatening public health in Brazil. Control actions were conducted in the area of the outbreak with the support of the army by helicopter, as it was difficult to access the area in the middle of the forest. Other countries in the Amazon Region reported outbreaks from bats in this period, mostly Peru [25,47].

To better understand the problem and suggest appropriate control actions, a new advisory group was convened by the Ministry of Health. Another transdisciplinary and intersectoral group was created, including the participation of state coordinators. Epidemiological indicators were developed to analyze the risk of rabies occurrence through viruses transmitted by hematophagous bats in different scenarios. This methodology included indicators related to the presence of the vampire bat and evidence of the circulation of the virus in animals and people bitten by bats, as well changes in the productive process such as deforestation and gold mining, as evidence of alteration in the environment and probably affecting vampire bat behavior. Four levels of risk scenarios were developed, and recommendations for actions for each were suggested; for the first time in Brazil, pre-exposure prophylaxis for high-risk remote areas was suggested (Table 3). This activity in 1991 demonstrated the integrated vision of the Program, as it recommended intersectoral activities among health and agriculture and considered environmental changes as a risk that may affect the spillover of the virus to people. Guidelines were made with this information, and joint training with the participation of the health and agriculture sector was developed. 

In this period, a mathematical model was developed to study different control measures (pre- and post-exposure vaccination, control of bat populations, or a combination of the two) in order to analyze which effort would work best in terms of reducing the risk of human rabies in remote areas where people are constantly at risk of being bitten by hematophagous bats (*Desmodus rotundus*) [25,48]. The conceptual and mathematical model for this research can be found in Appendix A.

A cross-sectional study was completed in a remote gold-mining village to obtain four of the Parámeters for this model; it was found that 41% of the village people interviewed had been attacked by a bat at least once [25,49]. The potential force of infection (per capita rate at which susceptible individuals acquire infection) of human rabies by the virus transmitted by the common hematophagous bat was estimated in the event that the rabies virus was introduced to a colony of bats close to a village with a high rate of human bites. It was estimated to be 0.0096 per person per year, meaning a risk of almost 1 case (0.96) per 100 people living in an area with an incidence of bites that was found in the study village [48]. This estimation was eventually confirmed ten years later in another outbreak in 2004 in Brazil near the area of the cross-sectional study [50]. The results of the final mathematical model suggested that a combination of bat population control and pre-exposure prophylaxis would better reduce that risk in a similar situation [25]. 

In 1998, two more human cases as a result of virus transmitted by *Callithrix jacchus* were reported, both in the state of Ceará in the Northeast Region. From samples taken from these cases, the rabies virus was able to be isolated and genetically identified. With the detection of a new rabies virus variant related to *Callithrix* sp., non-human primates stood out as “new” and important reservoirs of the rabies virus to humans [29]. The new *Callithrix* sp. variant was compared with isolated rabies virus dog variants, variants maintained by American wild terrestrial reservoirs, and variants from endemic cycles maintained by bats, and this marmoset species was considered as the source of exposure and maintained an independent cycle of a unique rabies virus variant [51]. These animals are commonly captured in the wild and brought home as a pet. Before the awareness campaign, most people were not aware of the risk. After the detection of *Callithrix* sp. as a potential source of rabies, a new policy was created to discourage domesticating wild animals; the need to conduct surveillance of epizootics in non-human primates and carry out specific studies was also noted.

These findings are also noteworthy, as non-human primates have rarely been reported with clinical rabies in the wild and have only sporadically been involved in cases of human exposure to the rabies virus [51,52,53]. Endemic in the Brazilian Northeast Region, this white-tufted marmoset is considered the only primate reservoir of the rabies virus and has also been associated with sporadic and unpredictable human deaths (10.63%) in Brazil [52,53]. De Sousa found a close evolutionary relationship between rabies viruses circulating in bats and variants hosted in white-tufted marmosets associated with human deaths in Brazil [54]. In this period, human cases in neighboring states were detected, and the virus was identified to be of the same new strain isolated from *Callithrix jacchus*.

Considering human cases mediated by dogs, 54% of the cases were reduced in this period (50 cases in 1990:23 cases in 1999). The same was not observed in dog cases; canine rabies went from 823 in 1990 to 1231 in 1999 (Figure 2). This could be a reflection of the structural changes in the way that dog vaccination campaigns were developed; first, it was the state responsible for this activity; however, with the new Unified Health System (SUS), the municipalities were responsible for developing the rabies vaccination campaigns, and this took a few years to adjust. However, the PEP continued to be offered free of charge in all health systems countrywide.

#### 3.1.4. Period D, from 2000 to 2010—The Shift between Cases Mediated by Virus in Dogs to Those Transmitted by Wild Animals

In this decade, the switch between dog-mediated cases and those transmitted by wild animals was clear. In 2000, 24 human cases of the rabies virus transmitted by dogs were registered. However, in 2008, for the first time, Brazil achieved the goal of zero dog-mediated human rabies cases, and the decade ended with only two dog-mediated cases.

On the other hand, large outbreaks as a result of rabies virus transmitted by vampire bats occurred in remote areas of the Legal Amazon Region, with 22 cases in 2004, most of them in Pará state (21 cases), and 42 human cases during 2005, most in two states (24 in Maranhão and 17 in Pará state). An emergency plan was created, and a large task force combining health and agriculture sectors (national, state, and municipal levels) was created to control the outbreaks of human rabies transmitted by common vampire bats in remote areas. This plan included a multidisciplinary team involving professionals from different areas (physicians, veterinarians, biologists, nurses, psychologists, social workers, community agents and leaders, and assistants’ nurses, among others), as well as from different sectors, health, agriculture, environmental, military, and others. Among the activities performed were (a) delivering PEP developed by the health sector locally for people exposure and active surveillance of people bitten by bats; (b) control of bat population developed by the agriculture sector; (c) rabies vaccination campaign of dogs and cats in urban and rural area; (d) educational materials and awareness about the risk and need of PEP were a joint activity; and (e) distribution of food supplies for the families that moved temporarily from remote areas to stay closer to the health team to receive PEP. The multidisciplinary and intersectoral teams spent weeks in remote areas working jointly to control these outbreaks and avoid more human fatalities, as an example of the One Health approach.

During this period, most of Brazil’s municipalities (at that time, 5560) had PEP available. An average of 500,000 people were attended to at a healthcare facility annually after being bitten by an animal. In 2003, Brazil changed the human vaccine to one produced in cell culture, and the PEP continues to be free-of-charge [55].

Annually, around 23,000 samples of dogs were sent to perform diagnostic tests for the rabies virus (around 0.14% of samples of the dog population) during Period D [46]. Most of these animals were hit by a car on highways or died in veterinary clinics, as the health authorities encouraged them to be brought to a laboratory for surveillance purposes. The country continues to use the Parámeter of testing 0.2% of samples from the dog population for the rabies virus, as established in the 1980s.

During this period, most states performed mass dog vaccination campaigns (except the South Region that vaccinated international borders). At that time, the canine vaccine used in Brazil was produced in brain tissue (Fuenzalida and Palacios), but in 2008, cell culture vaccines for dogs started to be introduced. In Period D, coverage averaged around 86% (81% to 94%), and approximately 21 million animals were vaccinated (82% dogs) [55]. Higher-risk areas were vaccinated twice during the year, with around 25 million dogs vaccinated annually.

The number of canine cases dropped significantly from 921 cases in 2000 to 26 in 2009, a reduction of 97.2% (Figure 2). However, in 2006, the AgV1 variant in dogs was detected in the state of Mato Grosso do Sul, bordering Bolivia [55].

In this period, rabies became part of the WHO global plan to combat neglected diseases and eventually with the goal of zero dog-mediated human cases [56]. In 2004, a study developed by the PAHO with the countries’ participation demonstrated that there was a reduction of around 90% of human cases transmitted by dogs as well as canine cases between 1982 and 2003 [12,47]. To receive political support for rabies elimination, the PAHO study was presented at RIMSA, an intersectoral high-level meeting organized periodically by the PAHO in 2005 (Reunión Interamericana, a Nivel Ministerial, en Salud y Agricultura—RIMSA 14, 2005) and included a PAHO resolution to eliminate neglected diseases approved by all countries in the region in 2009, with a target of zero dog-mediated human cases [11,56]. In this period, the “Alliance of Rabies Control” and the World Rabies Day were created, and these gave a vital push for rabies visibility and control actions.

#### 3.1.5. Period E, from 2010 to 2022—Almost Eliminating Human Cases Mediated by Dogs and Increasing the Wild Canid Cases

In this last period, for the first time, no human cases of rabies (by any transmitter animal) were reported in 2014; the number of human cases mediated by dogs was low (8 cases in total) and, for eight years, no dog-mediated cases were reported (in 2014 and 2016 to 2022), approaching the PAHO and WHO goal.

However, an outbreak of rabies in humans by a virus transmitted by bats occurred in the state of Pará and also in the Legal Amazon Region, with 10 cases in 2018. In 2019, after this outbreak as a result of virus transmitted by bats in remote areas, the government started a pilot project for pre-exposure vaccination in high-risk areas in the Amazon Region. This pilot project vaccinated with pre-exposure prophylaxis, according to WHO guidelines, a riverside population of around 3000 living in remote areas with difficult access to healthcare centers. This pilot project was carried out in partnership between the Ministry of Health, the PAHO, and the State Department of Pará. Pre-exposure has been a part of WHO recommendations for high-risk groups, such as veterinarians, for many years (WHO 2010). More recently, the WHO also recommended pre-exposure vaccination to populations living in rabies-endemic areas where vampire bat rabies is known to be present [2]. Another initiative in Peru was also developed to deliver pre-exposure prophylaxis for people living in remote areas and at risk of the rabies virus transmitted by vampire bats. More recently, a systematic review was conducted to analyze the safety and immunogenicity of pre-exposure rabies prophylaxis and recommended when PEP is limited or delayed [57].

During this period, a total of 24 human cases through viruses transmitted by bats were reported across the country. Another outbreak also transmitted by bats occurred in an indigenous reservation in 2022, with four children’s cases occurring on the border of Minas Gerais and Bahia state. This was the first time that a rabies outbreak among indigenous people was reported. It was not possible to confirm the mode of virus transmission; however, through the epidemiological investigation, it was suspected that exposure to the rabies virus occurred in a single event by bats. All residents received pre-exposure prophylaxis with two doses of the vaccine, and health education actions were carried out in the territory. It should be mentioned that other species of bats could also be detected with the AgV3 variant, typically of hematophagous bats [58]. This could be the case of this outbreak in the indigenous reservation, in which the antigenic and genetic research of two samples of the virus were identified as AgV3; however, the epidemiological investigation could not confirm the transmission mode. In the epidemiological investigation, it was found that the injuries present in two children were compatible with defensive bites from a non-hematophagous bat.

Another wild animal, *Cerdocyon thous* (bush dog), increased its importance as a source of the rabies virus as a new potential spillover to humans, threatening public health during this period. Since then, two human cases were reported with a new variant of this wild canid, and more than 50 canine cases with the *Cerdocyon thous* variant were detected (Appendix A). Rabies in wild canids currently represents an important cycle of transmission of the rabies virus in Northeastern Brazil and has been associated with a specific variant of the rabies virus, the *Cerdocyon thous* variant [59,60,61,62].

In this study’s last period, the dog cases by variant *Cerdocyon thous* were more than three times higher than those by variant AgV2, characteristic of dog rabies [55,59,62]. Rabies in wild canids currently also represents an important cycle of the virus transmission in Northeast Brazil, mainly in the states of Ceará, Bahia, and Pernambuco. Since the 1980s, rabies cases by these species have been frequent, with rabies virus transmission occurring in dogs, cats, and humans [63].

*Callithrix* sp. transmitted four cases; five cases were transmitted by cats (four of them with the variant AgV3 of hematophagous bat). It is suggested that the different lineages of the rabies virus may be related to the biological diversity of ecosystems and the specific fauna in the Northeast Region of Brazil [62,64,65]. In this manner, humans directly interfere in the ecosystem and in the spread of viruses [62]. Spillover events into the human population remain under-explored; these findings underline the need to track multiple rabies virus variants, which may pose a threat to both humans and animals and, as such, constitute a public health concern.

From 2010 to 2022, 391 dog cases, most of them in the Northeast Region (71.61%), were reported to the surveillance system. In 2015, an outbreak of canine rabies of variant AgV1 occurred in the state of Mato Grosso do Sul, which borders Bolivia. Two municipalities reported a large number (72) of dog cases (Appendix A).

On control actions during this period, an average of 672,183 received PEP; most of them (81%) were attacked by a dog in urban areas; 8.4% received a vaccine and serum [66]. In 2018, the dog vaccination coverage was 82% [67]. During 2019, the strategy was more directed at outbreak control, vaccinating along the international border [67].

#### Rabies Virus Variants in Brazil

For this period, the human cases reported are available at the municipal level, and it was possible to analyze the genetic variant of the rabies virus as part of the regular surveillance; the variant was detected among most of the cases (91.1%). Currently, the National Reference Laboratory for the Ministry of Health surveillance system uses genetic sequencing to identify the rabies virus variants, among other techniques.

Among the 45 human cases reported in this period, 25 (55.5%) samples were identified as AgV3, related to hematophagous bats (including the four cases by cats). The AgV2 variant was detected in eight human cases, seven mediated by dogs and one by cats. The AgV1 variant, also related to rabies in dogs, that is close to being eliminated in Brazil, was detected in 2015 in one human case during the outbreak cited above. Four human cases of *Callithrix jacchus* variant and two of *Cerdocyon thous* were also found (Table 4).

From 2014 onward, the analysis of the genetic variant of the rabies virus found in dog cases became part of the surveillance system. From 2014 to 2022, 185 canine cases were reported, and the variant was identified in 93.51% of cases (Appendix A). Figure 3 presents the human and dog rabies cases by variant detected over biome, using the One Health approach.

Different lineages of the rabies virus may be related to the biological diversity of ecosystems and the fauna found specifically in Northeast Brazil [62,64,68]. The relevance of classifying rabies viruses genetically and molecularly is also related to studies on the adequacy of the rabies vaccine, to be used both as a prophylaxis before and after a potential infection. Therefore, as humans directly interfere in the ecosystem, they interfere in the spread of viruses [62]. The appearance of variants in a new host should be surveilled for modifications/mutations in order to monitor the adequacy of the rabies vaccine as [69] Poch et al. (1990) and [70] Ito et al. (2001) have found that small mutations, sometimes even of a single amino acid, in certain regions can alter pathogenicity and virulence. These findings underline the need to track multiple rabies virus variants that may pose a threat to both humans and animals, especially as spillover events to humans remain underexplored.

Genetic sequencing is being used to study the diversity of the rabies virus from domestic and wild animals identified in different regions of Brazil (North, Northeast, Southeast, South, and Central-West); this provides a better understanding of rabies occurence in the country.

Wanderler et al. (1993) [71] considered that there would be a main host species for each region and that virus transmission chains are the main factors capable of influencing the diffusion between distinct species. When a given population remained in greater isolation, the strain of the rabies virus present would be differentiated by natural selection within the viral population in adaptation to the host. Mattos et al. (2000) [72] also reported different strains of the rabies virus in Latin America, with their characterization performed by molecular studies. These authors attribute the diversity found mainly to the main host that maintains the viral population, with eventual passage to other animal species. They suggest that marketing wild animals as pets to areas far removed from where they were caught could further spread the virus.

Romijn et al. (2003) [73] carried out an epidemic–geographic rabies study by analyzing central nervous tissue for the rabies virus. Lyssavirus RNA, detected by a nested PCR assay, was submitted to sequencing of amplified rabies virus nucleoprotein encoding segments and resulted in the formation of clusters corresponding to samples originating from cattle and equines from the same hydrographic basin. Lyssavirus strains of bat origin that were related genomically were found in each cluster, most likely because of the role of the bat in the epidemiology of herbivore rabies.

### 3.2. Part 2—Exploring Possible Drivers during Period D (2010 to 2022)

#### 3.2.1. Describing the Human Cases in the Country over Selected Possible Drivers

To approach the One Health vision for the occurence of rabies in Brazil, a few variables were selected to describe the distribution of human cases in the entire country. These variables and other possible drivers were examined statistically only in the Northeast Region. 

Most of the human cases (64.44%) occurred in the tropical and subtropical moist broadleaf forest (TSMBF) biome, a larger major habitat type in Brazil, and 20% in the major habitat type of deserts and xeric shrublands (DXS), predominant in the Northeast Region (Figure 4). The majority of cases (75.55%) took place in municipalities with a mean temperature higher than 25 °C (Figure 5). The findings suggest that the majority of cases (75.56%) took place in municipalities with less than 50,000 habitants (Table 4); 57.78% of the cases occurred in municipalities with a demographic density lower than the national average (23.8 habitants by square feet) (Figure 6). Half of the cases (51%) occurred in municipalities with some of the lowest GDP per capita rates in the country (Figure 6). Among the 29 municipalities with rabies cases, 27 presented a Gini index higher than 0.5, which is considered a high-income inequality (Table 4).

The majority of cases transmitted by bats (20/24) occurred in municipalities where the major biome was TSMBF, and most cases (95.83%) were considered rural (Table 4). Human cases mediated by dogs were present in different major habitat types, most of them in urban areas (62.5%). The cases transmitted by non-human primates (Callithrix jacchus) occurred all in the Northeast Region; three of the four cases are in the biome DXS, and these cases were in both zones. The two wild canines (Cerdocyon thous) cases were also in the Northeast Region; one was in the desert biome, and the other was in the TSMBF, both being rural.

#### 3.2.2. Descriptive and Statistical Analysis of the Northeast Region

As mentioned in the methodology, this region historically represents a major challenge to controlling human rabies cases in Brazil. Several possible demographic, socioeconomic, and environmental drivers were described and examined through a regression model. 

During Period D, 19 human cases were reported in 17 municipalities in the region; 7 (36.8%) of them were dog-mediated, 4 by Callithrix sp. (21.1%), 3 by bats (15.8%), 3 by cats (15.8%) and 2 by wild canids (10.5%). The majority (63.16%) of the human cases were transmitted by species other than canines. The virus of the two cat-transmitted cases was a wild species. 

In terms of possible environmental drivers, the 17 municipalities with human cases were distributed in three different major habitat types in this region (9 = DXS, 5 = TSMBF, and 3 = M); this can be further broken down to tropical environments (eight municipalities) and non-tropical environments (nine). In both biome types, the mean temperature ranges from 27.8 Celsius to 19.5, and the majority (84.21%) of the cases occurred in municipalities with temperatures higher than 25 Celsius. Precipitation ranges from 2645 to 371 (Appendix A).

Possible socioeconomic drivers were explored, including GDP per capita (range: BRL 4624.55 to 220,212.95) and Gini index (range: 0.7972 to 0.3684). As previously mentioned, the Northeast Region has a lower GDP per capita compared with other regions. Most (15/17) of the municipalities with cases were in the 0.5 or higher Gini index, which means high-income inequalities (Table 4). It also examined demographic density that ranged from 9427.22 habitants by Km^2^ in the states’ capitals to 0.87 habitants by Km^2^ in rural municipalities (Appendix A).

The covariables used as a proxy for access to PEP were primary healthcare coverage; this variable presented an average during the period that ranged from 45.5% to 100% of coverage, and the majority of cases were in municipalities with lower coverage. The other covariable was dog vaccination coverage, analyzed as an average for the period and ranges from 18.48% to 100% coverage. 

For the statistical analysis to estimate the association of human rabies cases with possible drivers using logistic regression, first, a univariate analysis was run with eight possible risk factors and two covariables. As a result of the univariate, it was found that temperature was associated and precipitation was borderline, and the primary healthcare coverage suggested a protective effect; the dog vaccination coverage variable was also borderline (Appendix A). In the final multivariable model, two variables were significant: temperature (OR = 1.739; CI_95%_ = 1.181–2.744), suggesting higher temperature increases the risk for rabies, and primary healthcare coverage (OR = 0.947; CI_95%_ = 0.915–0.987), suggesting the municipalities with higher coverage of Family Health Teams present a lower risk of human rabies cases.

The results of part II of this study related to the drivers for the occurrence of rabies in the Northeast of Brazil, which is the most affected region in Brazil (53% of the total cases) and has the four genotypes of the rabies virus, AGV2 in dogs, AgV3 in bats, Ag *Callithrix* in non-human primates, and Ag *Cerdocyon thous* in wild canids, indicated the association of temperature with the presence of human cases of rabies. The effect of temperature on the induction of interferons by virus infection has already been studied by Postic et al. in 1966 [74]. Moriyama and Ichinohe (2019) showed that the exposure of mice to the high ambient temperature of 36 °C impaired adaptive immune responses against infection with some viral pathogens [75]. Their findings uncover an unexpected mechanism by which ambient temperature may interfere with virus-specific adaptive immune responses. Several authors suggest that the elevation of global temperatures increases the transmission of viruses between different animals [76,77,78,79]. In fact, climate change may expand the geographical distribution of several pathogens, and the effects of environmental temperature may affect the host’s defense against viral infection. Scholars have been analyzing through mathematical models the possible expansion of risk areas for rabies transmitted by vampire bats due to increasing temperatures, creating more suitable environments for the distribution of the *Desmodus rotundus* [80,81].

At least 10,000 virus species have the ability to infect humans, but at present, the vast majority are circulating silently in wild mammals [77,82]. However, changes in climate and land use will lead to opportunities for viral sharing among previously geographically isolated species of wildlife [78,79]. In some cases, this will facilitate zoonotic spillover—a mechanistic link between global environmental change and disease emergence [76]. Carlson et al. [76] simulate potential hotspots of future viral sharing using a phylogeographical model of the mammal–virus network and projections of geographical range shifts for 3139 mammal species under climate change and land-use scenarios for the year 2070. They predict that species will aggregate in new combinations at high elevations, in biodiversity hotspots, and in areas of high human population density, such as in Asia and Africa, causing the cross-species transmission of their associated viruses an estimated 4000 times. Owing to their unique dispersal ability, bats account for most of the novel viral sharing and are likely to share viruses along evolutionary pathways that will facilitate future emergence in humans. Their findings highlight an urgent need to pair viral surveillance and discovery efforts with biodiversity surveys tracking the range shifts of species, especially in tropical regions that contain the most zoonoses and are experiencing rapid warming.

Our study also suggests that municipalities with higher primary health care coverage, in this case by the SUS Family Health Program, have a lower risk of human rabies cases. This variable was used as a proxy for access to PEP, as it is one of the SUS facilities providing this vaccine. Through an analysis of the National Health Survey conducted in 2019, Giovanella et al. (2021) [83] highlighted the importance of the coverage of the Family Health Program as an equitable health policy in the country and also reported the availability of regular demand services being higher among those registered in family health units. However, in the Amazon, there are communities characterized, among others, by the unequal distribution of resources and equipment, health care, and the development of surveillance activities, in addition to the possibility of indicated prophylactic conduct performed in an inappropriate way that could be interfering in PEP [66,84]. Another study pointed out a limitation to the appropriate use of PEP: the existence of a “gap” in the knowledge of health professionals who care for people bitten by animals that could transmit the rabies virus [85]. The training of health care personnel is still essential.

According to the SUS strategy, the municipalities’ Family Health Team is made up of a doctor, nurse, technician, or nursing assistant and community health agents; it may also include a dentist and dental hygiene technician, among other professionals [32]. However, this multidisciplinary Family Health Team could include veterinary doctors and zootechnics that may work with zoonosis control, among other activities [86].

The Ministry of Health annually establishes the goals and actions to be developed by states and municipalities, respecting their specificities based on the analysis of the epidemiological situation of each disease. Among the activities related to rabies are strengthening surveillance at the municipal level, including notification, investigation, and laboratory confirmation, and providing PEP. Also, as part of SUS, people who have not finished the PEP receive multiple attempts to follow-up from community health agents and sometimes visits to observe the suspected aggressor animal by the health team, mostly when veterinarians are part of the local team. In this context, the control of canine rabies was included as a goal agreed with the states and municipalities to reduce human rabies cases. Dog vaccination campaigns and outbreak control are also part of SUS. The decentralization of the activities and the agreement among decision makers at the municipal level could be considered an important step in rabies control.

As a limitation of this study, it could be suggested that not all information related to rabies control in these fifty years could be recovered; Brazil is a large country with activities and decentralized data. Another possible limitation of this study is the use of aggregated data by municipality in the driver’s analysis (part II of the study), as ecological studies are commonly associated with ecological fallacy [87].

## 4. Conclusions

The Brazilian Rabies Program demonstrated an enormous effort to eliminate dog-mediated human cases and reduce the transmission of variant AgV1 and AgV2 among dogs; zero dog-mediated human cases were reported for several years, achieving the goal established by the PAHO in 2009 and by the WHO and the United Against Rabies collaboration [5,6,56] (Appendix A). The major activities developed in Brazil for these achievements include free access to PPE, mass dog vaccination campaigns, surveillance, and awareness. However, the challenges of the rabies virus transmitted by wild animals continue (hematophagous bats, non-human primates, and wild canids), and activities must remain. As presented in this study, from 2010 to 2022, human cases transmitted by wild animals represent around 80% of human rabies cases in Brazil, the opposite percentage compared to when the Program started, and in the last decade, around 50% of human rabies cases were transmitted by hematophagous bats.

On the driver’s analysis, the association of human cases with higher temperatures in the Northeast is a concern that may increase with climate change. Our study also suggests that municipalities with higher coverage by the SUS Family Health Program have a lower risk of human rabies cases, which implies access to free PEP in the public health system in the Northeast Region. However, in the Amazon, there are communities living in remote areas with difficult access to health care. To prevent human rabies cases transmitted by wild animals, it is essential to continue to offer free PEP, and for remote at-risk areas, continue the pre-exposure vaccination project, especially in the Amazon Region, as well as increase awareness.

As described in this study, from the beginning of the Brazilian Rabies Program, the joint work among professionals of different disciplines at all levels, from expert committees at the national level to the local level, as part of the Family Health Team working in close collaboration with the community agent, was fundamental. The surveillance system was integrated from the beginning and control actions, free of charge to the population, always include humans and animals, as well as collaboration with the agriculture sector and taking into consideration the environment and socioeconomic conditions. These characteristics of the Brazilian Rabies Program are vital to the achievements of dog-mediated rabies control in Brazil and to face the challenge of wild animal cases and could be considered as an example of the One Health initiative.

## Figures and Tables

**Figure 1 pathogens-12-01342-f001:**
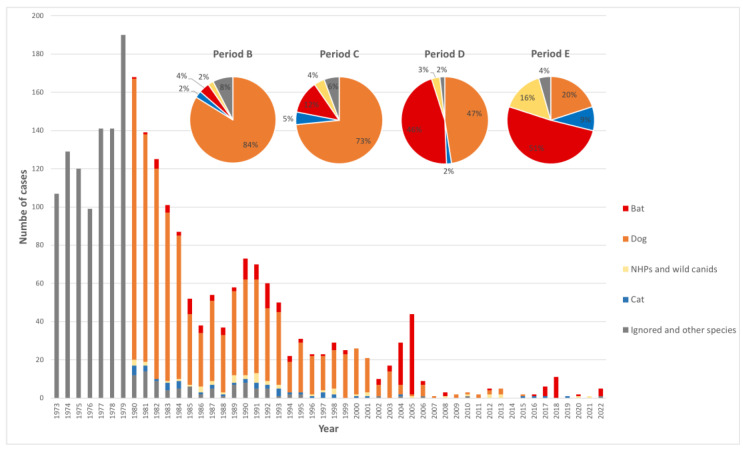
Number of cases of human rabies according to transmitter animal by year, and percentage of transmitting animal by period, Brazil, 1973–2022. Sources: Health Ministry of Brazil and other sources recovered by authors.

**Figure 2 pathogens-12-01342-f002:**
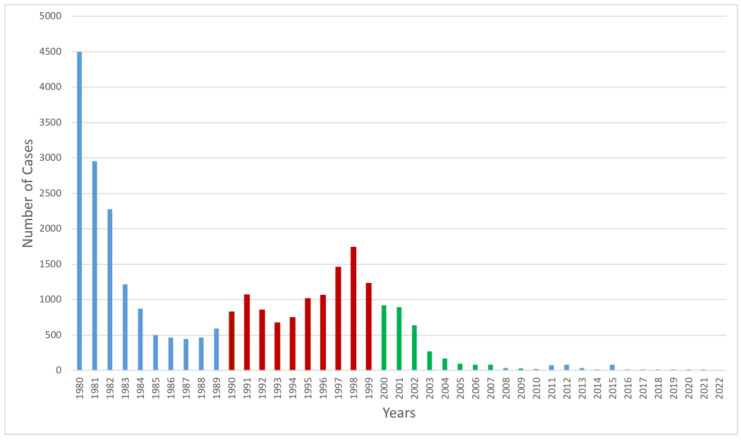
Number of canine cases of rabies, by year, Brazil, 1980–2022. Sources: Health Ministry of Brazil, 2023.

**Figure 3 pathogens-12-01342-f003:**
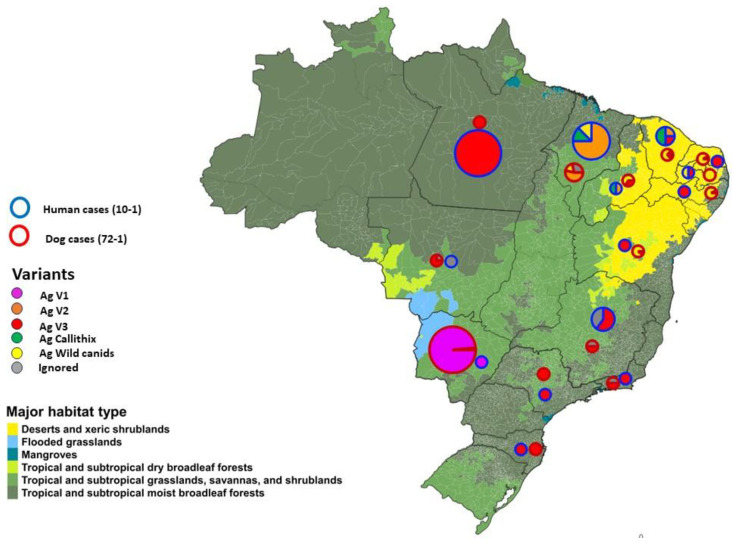
One Health map of human rabies by variant (2010–2022) and dog rabies by variant (2014–2022) and by state over biome, Brazil.

**Figure 4 pathogens-12-01342-f004:**
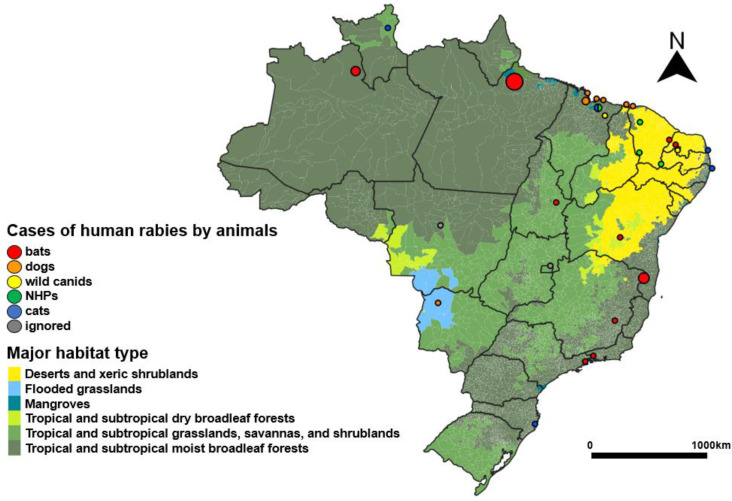
Cases of human rabies according to transmitting animal (dogs, bats, non-human primates, wild canids, cats, and ignored) by municipality over major habitat type, Brazil, 2010–2022.

**Figure 5 pathogens-12-01342-f005:**
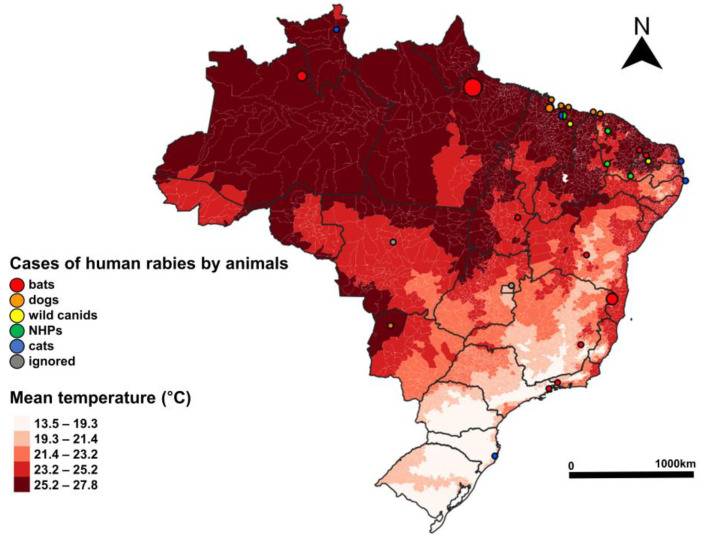
Cases of human rabies according to transmitting animal (dogs, bats, NHPs, wild canids, and cats) by municipality over temperature, Brazil, 2010–2022.

**Figure 6 pathogens-12-01342-f006:**
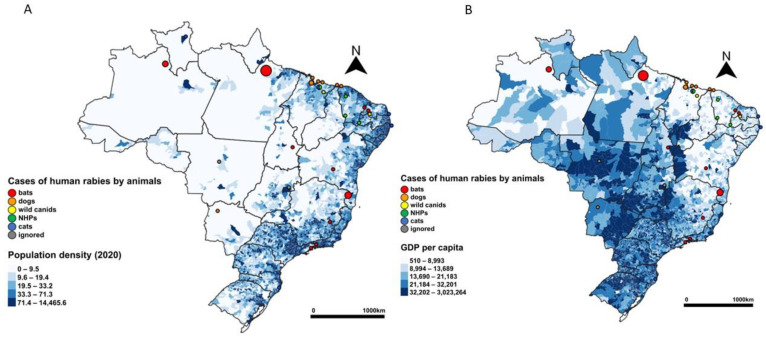
Cases of human rabies according to transmitting animal (dogs, bats, non-human primates, wild canids, cats, and ignored) by municipality over population density (**A**) and over GDP per capita (**B**), Brazil, 2010–2022.

**Table 1 pathogens-12-01342-t001:** Population, GDP per capita, and number of cases of human rabies, by period, region, and state, Brazil, 1980–2022.

Region	Population	GDP per Capita	1980–989 Period A	1990–999 Period B		2000–2009 Period C		2010–2022 Period D	Total	Percent (%)
State										
**North**	**18,672,591**	**25,608**	**132**		**86**		**61**	**15**	**294**	**19.71**
Acre	894,470	18.420	0		16		03	0	19	3.42
Amapá	861,773	21.432	7		01		0	0	8	1.27
Amazonas	4,207,714	27.573	31		06		02	03	42	2.82
Pará	8,690,745	24.847	73		35		45	10	163	10.92
Rondônia	1,796,460	28.722	21		21		09	0	51	3.42
Roraima	631,181	25.388	0		0		0	01	1	0.07
Tocantins	1,590,248	27.448	0		07		02	01	10	0.67
**Northeast**	**53,374,243**	**18,812**	**456**		**239**		**85**	**19**	**799**	**53.55**
Alagoas	3,351,543	18.858	71		26		05	0	102	6.84
Bahia	14,930,634	20.449	82		55		09	01	147	9.85
Ceará	9,187,103	18.168	94		29		13	04	140	9.38
Maranhão	7,114,598	15.028	50		55		48	08	161	10.79
Paráíba	4,039,277	17.402	22		14		0	02	38	2.55
Pernambuco	9,616,621	20.101	78		40		04	01	123	8.24
Piauí	3,281,480	17.185	26		12		03	02	43	2.88
Rio Grande do Norte	3,534,165	20.342	26		02		00	01	29	1.94
Sergipe	2,318,822	19.583	7		06		03	0	16	1.07
**Southeast**	**89,012,240**	**44,406**	**184**		**48**		**10**	**07**	**249**	**16.69**
Espírito Santo	4,064,052	34.066	07		07		02	0	16	1.07
Minas Gerais	21,292,666	32.067	65		35		06	05	111	7.44
Rio de Janeiro	17,366,189	43.408	59		0		01	01	61	4.09
São Paulo	46,289,333	51.365	53		06		01	01	61	4.09
**South**	**30,192,315**	**43,327**	**5**		**0**		**0**	**01**	**6**	**0.40**
Paráná	11,516,840	42.367	1		0		0	0	1	0.07
Rio Grande do Sul	11,422,973	41.228	1		0		0	0	3	0.20
Santa Catarina	7,252,502	48.159	3		0		0	01	2	0.13
**Central-West**	**16,504,303**	**47,942**	**96**		**38**		**07**	**03**	**144**	**9.65**
Federal District	3,055,149	87.016	0		00		00	01	1	0.07
Goiás	7,113,540	31.507	81		25		03	0	109	7.31
Mato Grosso	3,526,220	50.663	9		11		04	01	25	1.68
Mato Grosso do Sul	2,809,394	43.649	6		02		0	01	9	0.60
**Total**	**211,755,692**	**35.936**	**873**		**411**		**163**	**45**	**1492**	**100**

Sources: Cases Health Ministry of Brazil; Population and GDP per capita from IBGE, 2020.

**Table 2 pathogens-12-01342-t002:** Epidemiological tool used to analyze rabies virus transmitted by dogs created by a transdisciplinary committee during Period B.

Groups of Indicators	Indicators
**Epidemiological situation**	**Productive area:** Where there is a known active transmission of the rabies virus to people and/or domestic animals within the past three years. Indicators: Cases of rabies in people and/or domestic animals. **Silent area:** Area in which the information system is considered unreliable; the recommendation is to send 0.2% of annual samples from the estimated canine population for laboratory diagnosis of rabies. Indicators: Number of canine samples sent. **Vulnerability:** Probability of the introduction of a case to an area where no cases have been occurring. Indicators: Number of cases of any kind in neighboring municipalities; number of imported canine cases; and presence of infected vampire bats in the area. **Receptivity:** Probability of the occurrence of new cases of rabies after virus introduction into an area. Indicators: Dog vaccination coverage; outbreak control; and observation of aggressive animals.
**Control actions**	Relationship between health units with treatment and population. Immunoglobulin application. Percentage of people treated in relation to those assisted. Percentage of treatment abandonment. Percentage of observed aggressive animals. Vaccination coverage. Outbreak control. Percentage of samples sent for diagnosis in relation to the canine population. Animal care centers. Existence of hematophagous bat control efforts. Existence of diagnostic laboratory.
**Socioeconomic conditions**	Proportional mortality. Migration. Population coverage of safe water. Income up to two minimum wage salaries. Illiteracy.

**Table 3 pathogens-12-01342-t003:** Epidemiological indicators to analyze rabies virus transmitted by hematophagous bats created transdisciplinary committees during Period B.

	**Risk Situation**		**Actions to Be Developed**	
**Type I**	- Presence of vampire bats - Changes in the productive process/environment, such as deforestation and gold mining		- Surveillance of people bitten by bats - Surveillance of animals bitten/with rabies - Awareness/education	
**Type II**	- Presence of vampire bats - Changes in the productive process/environment - Evidence of rabies virus circulation	*Situation 1* With information system *Situation 2* Without information system	*To all* - Surveillance of people bitten by bats - Surveillance of animals bitten/with rabies - Awareness/education	*Situation 1* Request from the health care services negative notification of people bitten by bats *Situation 2* Active surveillance of people bitten by bats
**Type III**	- People bitten by bats - No evidence of rabies virus circulation	*Situation 1* With access to health care services *Situation 2* Without access to health care services With suspicion of virus circulation	*To all* - Control of the bat population - Awareness/education - Epidemiological investigation - Surveillance to confirm virus circulation	*Situation 1* PEP of people bitten by bats *Situation 2* PEP of people bitten by bats Pre-exposure vaccination for high-risk groups
**Type IV**	- People bitten by bats - Evidence of rabies virus circulation	*Situation 1* With access to health care services *Situation 2* Without access to health care services With suspicion of virus circulation	*To all* - Control of the bat population - Awareness/education - Epidemiological investigation - Joint actions by health and agriculture authorities	*Situation 1* PEP of people bitten by bats *Situation 2* PEP of people bitten by bats Pre-exposure vaccination for high-risk groups

**Table 4 pathogens-12-01342-t004:** Cases of human rabies by transmitter animal, population, variant, zone of the case, by first and second subnational level, Brazil, and major habitat type and Gini index of the municipality, Brazil, 2010–2022.

Region	Municipality (Population)	HR Bats	HR Dogs	HR Cats	HR NHP	HR Wild Canids	HR Ign	Variant	Zone	Major Habitat Type	Gini Index
State											
**North**		**14**	**0**	**1**	**0**	**0**	**0**				
Amazonas	Barcelos (27,638)	3	0	0	0	0	0	AgV3	R	TSMBF	0.7367
Pará	Melgaco (27,890)	10	0	0	0	0	0	AgV3	R	TSMBF	0.5537
Roraima	Boa Vista (419,652)	0	0	1	0	0	0	AgV3	U	TSGSS	0.5936
Tocantins	Ponte Alta do Tocantins (8116)	1	0	0	0	0	0	AgV3	R	TSGSS	0.6420
**Northeast**		**3**	**8**	**2**	**4**	**2**	**0**				
Bahia	Parámirim (21,695)	1	0	0	0	0	0	AgV3	R	TSMBF	0.5134
Ceará	Chaval (13,091)	0	1	0	0	0	0	AgV2	R	DXS	0.5062
	Ipu (42,058)	0	0	0	1	0	0	AgV Callithrix	R	DXS	0.5648
	Iracema (14,326)	1	0	0	0	0	0	AgV3	R	DXS	0.5562
	Jati (8130)	0	0	0	1	0	0	AgV Callithrix	R	DXS	0.4995
Maranhão	Chapadinha (80,195)	0	0	0	0	1	0	Wild canids	R	TSMBF	0.6019
	Humberto de Campos (28,932)	0	1	0	0	0	0	AgV2	U	TSMBF	0.6312
	Mirinzal (15,011)	0	1	0	0	0	0	AgV2	U	TSMBF	0.5160
	Paco do Lumiar (123,747)	0	1	0	0	0	0	AgV2	R	M	0.5067
	Sao Jose do Ribamar (179,028)	0	2	0	0	0	0	AgV2	R/U	TSMBF	0.5268
	Sao Luis	0	2	0	0	0	0	AgV2	U/U	TSMBF	0.6266
Paráíba	Catole do Rocha (30,684)	0	0	0	0	1	0	Wild canids	R	DXS	0.5046
	Jacarau (14,450)	0	0	1	0	0	0	AgV3	R	DXS	0.5499
Pernambuco	Recife (1,653,461)	0	0	1	0	0	0	AgV3	U	M	0.6894
Piauí	Parnaiba (153,482)	0	0	0	1	0	0	AgV Callithrix	U	DXS	0.5772
	Pio IX (18,459)	0	0	0	1	0	0	AgV Callithrix	R	DXS	0.5749
Rio Grande do Norte	Frutuoso Gomes (4041)	1	0	0	0	0	0	AgV3	R	DXS	0.4768
**Southeast**		**7**	**0**	**0**	**0**	**0**	**0**				
Minas Gerais	Bertopolis (4607)	4	0	0	0	0	0	(4)AgV3	R	TSMBF	0.5442
	Rio Casca (13,473)	1	0	0	0	0	0	AgV3	R	TSMBF	0.4741
Rio de Janeiro	Angra dos Reis (207,044)	1	0	0	0	0	0	AgV3	R	M	0.5293
São Paulo	Ubatuba (91,824)	1	0	0	0	0	0	AgV3	PU	TSMBF	0.5971
**South**		**0**	**0**	**1**	**0**	**0**	**0**				
Santa Catarina	Gravatal (11,577)	0	0	1	0	0	0	AgV3	R	TSMBF	0.3993
**Central-West**		**0**	**1**	**0**	**0**	**0**	**2**				
Federal District	Brasilia (3,055,149)	0	0	0	0	0	1	AgV3	U	TSGSS	0.6370
Mato Grosso	Tapurah (14,046)	0	0	0	0	0	1	AgV3	R	TSMBF	0.5586
Mato Grosso do Sul	Corumbá (112,058)	0	1	0	0	0	0	AgV1	U	FG	0.5589
**Total**		**24**	**9**	**4**	**4**	**2**	**2**				

Legend: major habitat type: tropical and subtropical moist broadleaf forests (TSMBF); tropical and subtropical grasslands, savannas, and shrublands (TSGSS); deserts and xeric shrublands (DXS); mangroves (M); flooded grasslands (FG); and cases of human rabies (HR). Zone: urban (U), rural (R), peri-urban (PU). Sources: Major habitat type, FAO; Gini index 2010, IBGE; and cases, Health Ministry of Brazil.

## Data Availability

The data used in this publication are available in tables in the manuscript and in Appendix A. Other open access data links are available in the references list and Appendix A.

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
