# Peer review of "Fifty Years of the National Rabies Control Program in Brazil under the One Health Perspective"

_pathogens, 2023, doi:10.3390/pathogens12111342_

Round 1

Reviewer 1 Report

Comments and Suggestions for Authors

Thank you for the opportunity to review.  This paper represents quite a lot of work and contains valuable information.  I have suggested edits that I think would be important to review and consider prior to publication.  All comments and suggestions are embedded in the document.  

Comments on the Quality of English Language

Edits, in part, reflect suggestions regarding clarification in certain areas which I think may be due to quality of English.  

Author Response

Reviewer #1

Thank you for the opportunity to review. This paper represents quite a lot of work and contains valuable information.  I have suggested edits that I think would be important to review and consider prior to publication.  All comments and suggestions are embedded in the document. 

Thank you for your comments, they are very appreciated!

Line 67 - The first sentence in this paragraph is confusing.  Do you mean: In 1973, the National Program of Rabies Prophylaxis in Brazil was created through and agreement among the Ministry of Health, Ministry of Agriculture, CME and PAHO?

Thank you for your comment; the sentence has been amended to “In 1973, the National Program of Rabies Prophylaxis in Brazil was created through and agreement among the Ministry of Health, Ministry of Agriculture, CEME (Central de Medicamentos – Government center to purchase vaccines and medications) and PAHO” as suggested.

Line 73 - The word "reported" here does not seem to make sense.  Do you instead mean to use the word "developed" or maybe "created" such that the sentence reads "Also, the notification system for rabies cases was created by the Ministry of Health and Ministry of Agriculture?

This sentence has been rewritten “The surveillance system was a joint effort from Ministry of Health and by the Ministry of Agriculture, both sent the rabies cases to one combined bulletin (human, dog, cat cases by Ministry of Health, livestock cases by Ministry of Agriculture, both reported wild animal cases).”

Line 76 - I think this sentence would read better as "The Ministry of Health coordinated the National Program; all State Health Departments implemented these activities at the first subnational level."

This sentence now reads as follows “The Ministry of Health coordinated the National Program; all State Health Departments implemented these activities at the first subnational level.”

Line 79 - A little confused by what you mean by "...and frequent examples of joint work among sectors until now could be described."  Do you mean that this paper is the first one to attempt describing the joint efforts of these sectors?

Thank you for your comment. We deleted the “until now” and include “in different administrative levels”.

Line 111 - How were the variants for which no panel by the CDC has been established identified and confirmed?  Does Brazil have a panel to identify these?  I ask since, if these variants are going to be called out and identified in terms of CDC technology to identify variants, was just wondering what technology Brazil has and if how these and the other variants are identified and confirmed. 

During the 90’s the CDC panel was used in Brazil, and the citation in this sentence was from the Ministry of Health reference; however, this technique is not in use anymore. The National Reference Laboratory for Rabies Surveillance that is in Pasteur Institute in Sao Paulo, is now using genetic sequencing and other techniques. We deleted the sentence about the CDC panel and left only the reference and included in page 16 the current technique in use in Brazil to identify the genetic variants.

Line 116 - should be "were" not was.

As suggested, the sentence at line 116 was changed to “It should be noted that, in Northeastern Brazil, these new variants of the rabies virus, 114 with no antigenic or genetic relationship to any known rabies variant found in bats or land 115 mammals in the Americas, were identified in association with cases of human rabies re- 116 ported in the state of Ceará, Brazil, in the 90’s transmitted via the white-tufted marmoset 117 (Callithrix jacchus), commonly called in Portuguese as sagüí” [29].”

Line 144 – Sometimes the word team when it is used in the context is capitalized and sometimes it is not. I don’t know that one is preferred over the other, I just think you should be consistent.

Thank you for your comment. All uses of “Team” in this context (Family Health Teams) have been capitalized and were corrected in Lines 144, 202, 755, and 802.

Line 163 - For this section 1. The first word in the phrases associated with the list of reasons for the design should not be capitalized.  Also, whenever you have a list in parentheses, you should either use e.g. if the list represents examples, but not an exhaustive list, or i.e. if what is listed does represent the only items/choices available

Thank you for your comment. The first word capitalization throughout this section has been removed and “i.e.,” was added to the list of variables.

Line 166 - I assume what you mean here is possible drivers for human rabies cases as opposed to the cases of animal rabies that cause human infection, but it would be good to clarify.

This has now been clarified – “For the last period (2010 to 2022), the Ministry of Health has human rabies cases available by the transmitting animal at the second subnational level (municipalities); we use this information to examine possible drivers for human rabies cases.”

Line 174 - Why is the Northeast region considered a challenge?  Is it because it's rural?  May be good to make a general statement about why it has been considered a challenge from the beginning of the program. 

A brief explanation has been included to address the challenges faced in the Northeast region – “b) the Northeast Region reported 53% of all human cases, and this Region is considered a challenge due to the higher number of human cases and its socioeconomic indicators since the creation of the Program.”

Line 206 – 209 - This paragraph speaks to results, not methods, so it should be put in the results section.

This paragraph was removed and included two sentences with Table S2 on page 5.

Line 274 - Is this word supposed to be Establishing?

Thank you! The title of Period A was corrected to read “Establishing and implementing the Program.”

Line 287 - Instead of "some", could this be quantified?  Typically in results sections, more definitive terms and numbers are used to characterize amounts with words like "some", "most", "many" , etc avoided or further substantiated by a number.  

Thank you for your comments. Uses of the phrases listed were reviewed and quantified when possible. Please find some examples listed here: “A cross-sectional study was completed in a remote gold-mining village to obtain four of the parameters for this model” and “This estimation was eventually confirmed ten years later in another outbreak in 2004 in Brazil near.”

Line 296 – If you use to word “large” it would be helpful to indicate how large it is compared to other years. In other words, large as in compared to what?

It was possible to quantify “large”- “In these first years of the surveillance system, 927 human cases were reported. A large number of dog cases (46,298 dog cases) were reported; however, the criteria for dog case notification were just being implemented, and the high number of dog cases in 1974 (12,475 dog cases) and 1975 (13,710), compared with an average of 4,669 during 1975 to 1979, could be a reflection of the adjustment of the reporting system (Table S3).”

Line 345, Table 2 – I think this is meant to be “known” not know.

Thank you for your comment. This correction was made in Table 2.

Line 430 - does this mean that there was a change in canine surveillance? It would be helpful to note an example of any "structural changes" in the Ministry of Health or what about the new SUS that may have influenced the increased reports of dog rabies. 

Thank you for your inquiry. This does not mean increased surveillance; the changes occurred were in which administrative level has the responsibility to do the campaigns. We clarified in the text.

“This could be a reflection of the structural changes in the way that the dog vaccination campaigns were developed; first was the states level responsible for this activity; however, with the new Unified Health System (SUS), the municipalities were responsible for developing the rabies vaccination campaigns and this took a few years to adjust.”

Line 513 - sometimes the word "period" is capitalized when referring to this window of time in the study/results and other times not.  Just be consistent.

Thank you. We reviewed all text and capitalized when referencing Period A through E.

Line 691 – 693 - What does "Supporting information" mean in this context...does this mean you are referring to Table S5?

We have changed the title of this reference to Table S5. Thank you for pointing this out.

Line 778 - Especially because this paper covers a wide range of time and issue, I think it would be good to have a table or chart as a quick reference timeline for major activities and events over time.  

Thank you, it is an excellent suggestion, we included as Figure S8.

Reviewer 2 Report

Comments and Suggestions for Authors

Firstly, I congratulate the authors for the article, as it has historical importance for rabies control in Brazil, describing and analyzing the 50 years of activities of the National Rabies Control Program and how the One Helath concept has already been applied in the program's actions from the beginning.

I think the manuscript is a little long and some information is repeated throughout the text, so the authors could revise it to reduce its length a little.

I'm not an expert in English, but in some cases I suggest that the authors can improve some expressions and make some changes, as suggested in the comments included in the pdf text.

Table 5 is missing from the manuscript or spelled incorrectly and should be Table S5 instead of 5. Please review this subject.

Another aspect that I suggest reviewing is in relation to the terms "antigenic variants" and "genetic variants". When we talk about an antigenic variant, it is an identification by monoclonal antibodies and the panel used in Brazil, from the CDC/Atlanta, does not identify the variants of marmosets and wild canids, as well as several species of non- hematophagous bats. These are genetic variants.

When we talk about these new variants, which were identified by genetic sequencing that revealed genetic diversity in the virus genome, we are talking about a genetic and not an antigenic variant. Therefore, these terms must be correctly used throughout the text.

I recommended that the article should be accepted with minor revisions and I made some suggrstions which are included in comments on the manuscript.

Author Response

Reviewer #2:

Firstly, I congratulate the authors for the article, as it has historical importance for rabies control in Brazil, describing and analyzing the 50 years of activities of the National Rabies Control Program and how the One Health concept has already been applied in the program's actions from the beginning.

Thank you for your comments; they are very appreciated!

I think the manuscript is a little long and some information is repeated throughout the text, so the authors could revise it to reduce its length a little.

We deleted sentences in different pages, and have rewritten paragraphs (see pages 2, 5, 9, 13, 14, 15 and 16).

I'm not an expert in English, but in some cases I suggest that the authors can improve some expressions and make some changes, as suggested in the comments included in the pdf text.

All text was reviewed and edited as necessary.

Table 5 is missing from the manuscript or spelled incorrectly and should be Table S5 instead of 5. Please review this subject.

Thank you for your comment, the table being referenced here has now been correctly cited as “Table 4,” included within the main text.

Another aspect that I suggest reviewing is in relation to the terms "antigenic variants" and "genetic variants". When we talk about an antigenic variant, it is an identification by monoclonal antibodies and the panel used in Brazil, from the CDC/Atlanta, does not identify the variants of marmosets and wild canids, as well as several species of non- hematophagous bats. These are genetic variants.

When we talk about these new variants, which were identified by genetic sequencing that revealed genetic diversity in the virus genome, we are talking about a genetic and not an antigenic variant. Therefore, these terms must be correctly used throughout the text.

Thank you for the clarification; this is not an easy topic, we reviewed the text. We deleted the CDC sentence in the introduction. Also, we included a sentence in page 16: “Currently the National Reference Laboratory for the Ministry of Health surveillance system uses genetic sequencing to identify the rabies virus variants, among other techniques.”

We also reviewed the sentence on page 19 - “Genetic sequencing is being used to study the diversity of rabies virus from domestic and wild animals identified in different regions of Brazil (North, Northeast, Southeast, South and Central-West), this provides a better understanding of rabies transmission in the country. ”

I recommended that the article should be accepted with minor revisions and I made some suggestions which are included in comments on the manuscript.

Thank you very much, we appreciate that!

Other comments in the text:

Line 15 – Unesp – So Paulo State University

Thank you, the title of this affiliation was changed to “São Paulo State University, Brazil (Ret.)”

Also, we have translated the other Brazilian institutions to English.

Line 75 – animals

Thank you, done.

Line 108 - It's better use genetic variant because antigenic variant refers to those detected by monoclonal antibodies and the variants from sagui, wild canids and non-hematophagous bats were detected by genetic studies by molecular characterization

Thank you we reviewed in the text.

Line 125 – animals

Thanks, done.

Line 284 - in the late 90s

According to the Ministry of Health, the year was 2003. Based on this, the sentence has been corrected and the date clarified. “In this period, nerve tissue vaccines were used (Fuenzalida & Palacios type); replaced by cell culture vaccines in 2003.”

Line 329 - 339 - I think this paragraph could be improved. It's a little confusing

Please find the amended paragraph below.

“This scientific advice group developed several indicators to evaluate the risk of rabies at the municipalities level, the main one was related to surveillance reliability. The “silent area” indicator suggested that a surveillance system is considered unreliable if it does not achieve the recommendation to send 0.2% of annual samples from the estimated canine population for laboratory diagnosis of rabies every year. This indicator was in use at all levels for many years and became part of the SUS indicators for rabies control activities. Later, this indicator was reevaluated and recommended at the region level as 0.1% [47, 12]. A tool was developed using these indicators and others divided into three groups: a) epidemiological situation with major focus on surveillance, b) control actions related to human and animals, and c) socioeconomic conditions (Table 2) [25].”

Line 330 - I think that it could be replaced by "which"

This paragraph was rewritten and no longer includes this phrase, thank you.

Line 355 – I think it would be better to – due to or related to, the same suggestion for line 4 of this paragraph

Thank you for your comment. We changed the phrase to “as a result.”

Line 399 - include: some years later

Thanks, we included that it was ten years later.

Line 400 - it's unclear what region, this region?, the same region?, Amazon region?

This has been clarified – “This estimation was eventually confirmed ten years later in another outbreak in 2004 in Brazil near the area of the cross-sectional study [50].”

Line 405 - identified should be better

As suggested, this word has been substituted with “identified.”

Line 442 - due to

This correction has been made, thank you. The sentence now reads “On the other hand, large outbreaks of rabies virus transmitted by vampire bats occurred in remote areas of the Legal Amazon region”.

Line 463 – an

This change has been made, thank you.

Line 465 – The beginning of the sentence is missing here

The sentence has now been completed and reads as follows – “Annually, around 23,000 samples of dogs were sent to perform diagnostic tests for rabies virus (around 0.14% of samples of the dog population) during Period D [47].”

Line 469 – add: campaigns, except

Additions were included in the sentence. It now states, “During this Period, most states performed mass dog vaccination campaigns (except the South Region that vaccinated international borders).”

Line 473 - This sentence could substitute the one lines 440 and 441.

Thank you, the sentence has been rewritten.

Line 491 - were created

Thank you for your comment. This correction was made.

Line 498 – eight

This change has been made, thank you.

Line 503 – I think the word “vaccinations” should be better

The authors changed the word to vaccination, as suggested.

Line 509 - PrEP to population exposed to bats was also recommended by WHO position paper, 2018.- https://iris.who.int/bitstream/handle/10665/272372/WER9316-201-219.pdf?sequence=1

The following sentence has been added to the manuscript, “More recently WHO also recommended pre-exposure vaccination to populations living in rabies endemic areas where vampire bat rabies is known to be present [2].”

Line 522 - FAHL et al, 2012 described AgV3 detected in Desmodus and Artibeus

FAHL, W. O.; CARNIELI JR, P.; CASTILHO, J. G.; CARRIERI, M. L.; KOTAIT, I.; IAMAMOTO, K.; OLIVEIRA, R. N.; BRANDÃO, P. E Desmodus rotundus and Artibeus spp. bats might presente distinct rabies virus lineages. Brazilian Journal of Infectious Diseases, v. 16, n. 6, p. 545-551, 2012. DOI:  HYPERLINK "https://doi.org/10.1016/j.bjid.2012.07.002"10.1016/j.bjid.2012.07.002

QUEIROZ et al, 2012  also detected AgV3 in Artibeus and three more species

QUEIROZ, L. H.; FAVORETTO, S. R.; CUNHA, E. M. S.; CAMPOS, A. C.A.; LOPES, M. C.; CARVALHO, C.; IAMAMOTO, K.; ARAUJO, D. B.; VENDITTI, L. L.; RIBEIRO, E. S. Rabies in southeast Brazil: A change in the epidemiological pattern. Archives of Virology, v. 157, n. 1, p. 93-105, 2012. DOIisponível em: 10.1007/s00705-011-1146-1

Thank you, we replaced the previous reference 58 for the reference above.

Line 572 - put at the end of the table title

Thank you for your comment. The title of Table 4 has been amended according to this suggestion.

Line 602-605 - antigeic differences are detected by monoclonal antibodies and the molecular techniques detects genetic diversity on the sequence of aminoacids

The sentence must be rewrite

Thank you, we did it.

Line 655 - Table is missing in the manuscript

Thank you for your comment, the table being referenced here has now been correctly cited as “Table 4,” included within the main text.

Line 691 – It’s not in the manuscript

The table has now been correctly labeled as Table 5.

Line 785 – animals

Thank you, done.

Reviewer 3 Report

Comments and Suggestions for Authors

In several instances you mention the potential for climate change to impact the distribution of rabies and other zoonotic diseases. You might want to consider including the findings of Hayes and Piaggio (2018) and Lee et al. 2012 on the potential future distribution of vampire bats.

Lee DN, Papes¸ M, Van Den Bussche RA (2012) Present and Potential Future Distribution of Common Vampire Bats in the Americas and the Associated Risk to Cattle. PLoS ONE 7(8): e42466. doi:10.1371/journal.pone.0042466

Hayes MA, Piaggio AJ (2018) Assessing the potential impacts of a changing climate on the distribution of a rabies virus vector. PLoS ONE 13(2): e0192887

Line 45: should call rabies a neglected tropical disease

Are human cases and vegetation type (as in Figure 4) and human cases and temperature (as in Figure 5) correlated?

Reference #19 is a different font or size than the others.

Comments on the Quality of English Language

There are a few minor syntax issues. Otherwise, language is fine.

Author Response

Reviewer #3:

Thank you for your comments; they are very appreciated!

In several instances you mention the potential for climate change to impact the distribution of rabies and other zoonotic diseases. You might want to consider including the findings of Hayes and Piaggio (2018) and Lee et al. 2012 on the potential future distribution of vampire bats.

Lee DN, Papes¸ M, Van Den Bussche RA (2012) Present and Potential Future Distribution of Common Vampire Bats in the Americas and the Associated Risk to Cattle. PLoS ONE 7(8): e42466. doi:10.1371/journal.pone.0042466

Hayes MA, Piaggio AJ (2018) Assessing the potential impacts of a changing climate on the distribution of a rabies virus vector. PLoS ONE 13(2): e0192887

Thank you for your suggestion, we included it in the text (page 22). We changed the references number from there.

Line 45: should call rabies a neglected tropical disease

Thank you for your comment. Line 45 now reads as follows: “The World Health Organization (WHO) currently classifies rabies as a neglected tropical disease (NTD).”

Are human cases and vegetation type (as in Figure 4) and human cases and temperature (as in Figure 5) correlated?

Thank you, this is a very good question. We described human case distribution of the variable major habitat type or biome and over temperature. Figure 4 and 5 are descriptives only.

According to the FAO description, there are 14 major habitat types and six of them present in Brazil. A broad definition of biome includes climatic elements, flora and fauna.

In the statistical analysis we use only tropical biome vs non-tropical biome. Also, temperature and precipitation as continuous variables. Biomes and temperature could be correlated in lower latitudes or altitudes: however, some dry biomes (sertao - estepes or deserts) in this particular geographic situation, the temperature could be high, but humidity is very low, hence they are not classified as tropical. 

In the exploratory univariate analysis, the biome was not significant and did not stay in the final multivariate analysis. However, temperature showed a clear association with municipalities with rabies cases in Brazil.

We included a sentence in the methodology (line 231) to clarify this question.

Reference #19 is a different font or size than the others.

Thank you, we have fixed this.